# Are open set classification methods effective on large-scale datasets?

**Ryne Roady**[1], **Tyler L. Hayes**[1], **Ronald Kemker**[1], **Ayesha Gonzales**[2], **Christopher Kanan**[1,3,4]*

**1** Rochester Institute of Technology, Rochester, NY, United States of America, **2** Case Western Reserve University, Cleveland, OH, United States of America, **3** Paige, New York, NY, United States of America, **4** Cornell Tech, New York, NY, United States of America

* rpr3697@rit.edu

**Data Availability Statement:** All relevant data are within the paper and its Supporting Information files.

**Funding:** TH and CK were supported during this work in part by DARPA/MTO Lifelong Learning

## Abstract

Supervised classification methods often assume the train and test data distributions are the same and that all classes in the test set are present in the training set. However, deployed classifiers often require the ability to recognize inputs from outside the training set as unknowns. This problem has been studied under multiple paradigms including out-of-distribution detection and open set recognition. For convolutional neural networks, there have been two major approaches: 1) inference methods to separate knowns from unknowns and 2) feature space regularization strategies to improve model robustness to novel inputs. Up to this point, there has been little attention to exploring the relationship between the two approaches and directly comparing performance on large-scale datasets that have more than a few dozen categories. Using the ImageNet ILSVRC-2012 large-scale classification dataset, we identify novel combinations of regularization and specialized inference methods that perform best across multiple open set classification problems of increasing difficulty level. We find that input perturbation and temperature scaling yield significantly better performance on large-scale datasets than other inference methods tested, regardless of the feature space regularization strategy. Conversely, we find that improving performance with advanced regularization schemes during training yields better performance when baseline inference techniques are used; however, when advanced inference methods are used to detect open set classes, the utility of these combersome training paradigms is less evident.

## Introduction

Convolutional neural networks (CNNs) work extremely well for many categorization tasks in computer vision involving high-resolution images [1, 2]. However, current benchmarks use closed datasets in which the train and test sets have the same classes. This is unrealistic for many real-world applications. It is impossible to account for every eventuality that a deployed classifier may observe, and eventually, it will encounter inputs that it has not been trained to recognize. Open set classification (OSC) is the ability for a classifier to reject a novel input from classes unseen during training rather than assigning it an incorrect label [3]. This

Machines program [W911NF-18-2-0263], AFOSR grant [FA9550-18-1-0121], and NSF award #1909696. Additionally, CK was employed by a commercial company, Paige, New York during the preparation of this work. This funder provided support in the form of salaries, but did not have any additional role in the study design, data collection and analysis, decision to publish, or preparation of the manuscript.

**Competing interests:** One author in this study, CK, was employed at a commercial company, Paige, New York during the preparation of this manuscript. This company played no role in the sponsorship, design, data collection and analysis, decision to publish, or preparation of the manuscript. This interest does not alter our adherence to PLOS ONE policies on sharing data and materials.

capability is particularly important for the development of 1) safety-critical software systems (e.g., medical applications, self-driving cars) and 2) lifelong learning agents that must automatically identify novel classes to be learned by the classifier [4–7].

For OSC in large-scale datasets, the major challenge is the presence of 'unknown unknowns' since the set of possible inputs outside of the training set is unbounded. Within recent machine learning literature, the OSC problem is highly related to a number of different applications including selective classification [8], classification with a reject option [9, 10], and out-of-distribution (OOD) detection [11–14]. For our study, the goal of an open set classifier is to correctly classify inputs that belong to the same distribution as the training set and to reject inputs that are outside of this distribution. This is a narrower definition than the broad application of OOD detection which is only concerned with finding a function to determine whether an input belongs to the training distribution and not concerned with the correct classification of samples which are in-distribution.

Additionally in recent literature, the differences between these names have indicated a degree of difference between the training set distribution and the evaluation set containing outlier samples. In classification with a reject option, the test distribution has the same categories as the training distribution; however, a classifier should reject inputs it cannot confidently classify. In OOD detection as selective classification, the outlier data used in test cases often comes from broadly different data distributions than the training set or in special cases from adversarially generated data designed to fool a classifier into non-sensical predictions [15, 16]. In open set recognition, a classification model is often tested on known classes and novel inputs from related classes not observed during training. Surprisingly, there has been little work comparing methods developed for each of these paradigms.

To organize our evaluations we seperate the strategies for OSC into two general approaches. The first is specialized inference mechanisms for determining if the input to a pre-trained CNN should be rejected. The second is to alter the CNN during learning so that it acquires more robust representations of known classes that reduce the probability of a sample from an unknown class being confused. This often takes the form of collapsing class conditional features in the deep feature space of CNNs.

Finally, the vast majority of prior work for OSC in image classification has focused on small, low-resolution datasets, e.g., MNIST and CIFAR-100. Deployed systems like autonomous vehicles, where outlier detection would be critical, often operate on images that have far greater resolution and experience environments with far more categories. It is not clear from previous work if existing methods will scale. In this paper we compare methods across open set classification paradigms on large-scale, high-resolution image datasets.

Our major contributions are:

- We organize OSC methods for CNNs into families and identify the relevant benefits and penalties of different methods.

- We are the first to directly compare inference methods and feature space regularization strategies for OOD and open set recognition to quantify the benefit gained from combining these techniques.

- We extensively compare combinations of inference and feature space methods, many of which have not been previously explored, on the ImageNet large-scale classification datset.

- Using a challenging open set image classification protocols, we find that the relative OSC performance benefit from specialized inference methods is consistent across different OOD types and feature space representations.

- We also find that the relative benefit from certain feature space regularization methods such as confidence loss with a background class is very dependent on the relationship between the background class and the in/out-of-distribution datasets.

# 1 Problem formulation

While OSC is related to uncertainty estimation [17] and model calibration [18], its function is to reject outlier inputs to the CNN. We formulate the problem as a variant of traditional multiclass classification where an input belongs to either one of the $K$ categories from the training data distribution or to an outlier/rejection category, which is denoted as the $K + 1$ category. Given a training set $D_{train} = \{(X_1, y_1), (X_2, y_2), \ldots, (X_n, y_n)\}$, where $X_i$ is the $i$-th training input tensor and $y_i \in C_{train} = \{1, 2, \ldots, K\}$ is its corresponding class label, the goal is to learn a classifier $F(X) = (f_1, \ldots, f_k)$, that correctly identifies the label of a known class and separates known from unknown examples:

$$
\hat{y} = \begin{cases} \text{argmax}_k \ F(X) & \text{if} \ S(X) \geq \delta \\ K + 1 & \text{if} \ S(X) < \delta \end{cases}
\tag{1}
$$

where $S(X)$ is an acceptance score function that determines whether the input belongs to the training data distribution and $\delta$ is a threshold.

For testing, the evaluation set contains samples from both the set of classes seen during training and additional unseen classes, i.e., $D_{test} = \{(X_1, y_1), (X_2, y_2), \ldots, (X_n, y_n)\}$, where $y_i \in (C_{train} \bigcup C_{unk})$ and $C_{unk}$ contains classes that are not observed during training.

# 2 Open set classification in CNNs

We have organized methods for OSC into two complementary families: 1) inference methods that create an explicit acceptance score function for separating novel inputs, and 2) regularization methods that alter the feature representations during training to better separate in-distribution and novel samples.

## 2.1 Inference methods

Inference methods use a pre-trained neural network to perform OSC, but modify how the network outputs are used. Using pre-trained networks is advantageous since no modifications to training need to be made to handle outlier samples, and the feature representations of pre-trained deep CNNs have been shown to generalize across many different image datasets [19]. We briefly descript the OSC inference methods below and summarize their acceptance score functions and inference complexity in Table 1.

Table 1. The studied inference methods for OOD detection. Inference complexity refers to the number of passes through a deep CNN (forward and backward) during inference.

| CLASSIFICATION METHOD | ACCEPTANCE SCORE FUNCTION | INFERENCE COMPLEXITY |
|---|---|---|
| $\tau$-Softmax [11] | Simple Threshold | 1 |
| DOC [20] | Per-Class Threshold | 1 |
| ODIN [12] | Temp Adjusted Threshold | 3 |
| OpenMax [21] | Per-Class EVT Rescaling | 1 |
| One-Class SVM [22] | SVM Score | 1 |
| Mahalanobis [14] | Generative-Distance Metric | 3 |

**2.1.1 Confidence thresholding.** The simplest approach to OSC in deep CNN classification models is thresholding the output of a model after normalizing by a softmax activation function, thus producing a probabilistic confidence estimate among the known classes. For multi-class classifiers, the softmax layer assumes mutually exclusive categories, and in an ideal scenario would produce a uniform posterior prediction for a novel sample. Unfortunately, this ideal scenario does not occur in practice and serves as a poor estimate for uncertainty [23, 24]. Still, the max output of the softmax layer has been shown to follow a different distribution for outlier samples than in-distribution samples drawn from the known classes [11]. In our experiments, we refer to this thresholding on the confidence output of the model as $\tau$-Softmax.

Confidence thresholding can be further improved as a means of detecting open set classes by improving the model calibration during inference. Techniques for improving classification model calibration have been extensively studied [18, 25]; however, the Out-of-Distribution Image Detection in Neural Networks (ODIN) technique [12] was the first to extend these methods to exclusively improve OOD detection performance. This is accomplished by simultaneously temperature scaling the activation of the network prior to softmax output and applying a small input perturbation based on the gradient of this temperature adjusted softmax output. In this application, the sign of the gradient is used to enhance the probability of inputs that are in-distribution while minimally adjusting the output of outlier samples.

Additional work on confidence thresholding for OSC has focused on establishing per-class thresholds rather than a global threshold for rejection. One of the first methods to employ this strategy in deep neural networks was the Deep Open Classification (DOC) model [20], which alters a typical multi-class CNN architecture by replacing the softmax activation of the final layer with a one-vs-rest layer containing $K$ sigmoid functions for the $K$ classes seen during training. The sigmoid activation helps to avoid the normalization properties of the softmax activation and creates more discriminative per-class thresholds. A threshold, $k_i$, is then established for each class by treating each example where $y = k_i$ as a positive example and all samples where $y \neq k_i$ as negative examples. During inference, if all outputs from the sigmoid activations are less than the respective per-class thresholds, then the sample is rejected. For our evaluations, we separate this per-class thresholding strategy from the one-vs-rest model training strategy to isolate the benefits of the inference strategy from the benefits of training using a binary cross-entropy loss function.

**2.1.2 Distance metrics.** Outlier detection can also be done using a variety of distance-based metrics in deep feature space. Following the formulation of Knorr and Ng [26], a number of distance-based methods [27–30] have been developed based on global and local density estimation by computing the distance between a sample and the underlying data distribution including the Nearest Class Mean and Nearest Non-Outlier [31] metrics.

While Euclidean distance metrics have been typically used in the deep feature space of CNNs for OSC [32, 33], they often fail in high-dimensional feature spaces containing many classes. To mitigate this issue, the most successful recent method for OSC in CNNs uses a nearest class Mahalanobis metric [14] across the feature space of mulitple layers within the network. This method requires a seperate set of class means and a tied covariance matrix to be calculated (learned) at each layer within a network. It then requires that a linear classifier be learned through cross-validation for the metrics across different layers to be combined into a single acceptance score function. Interestingly, the utility of the Mahalanobis metic in deep feature space of pre-trained networks is justified by showing that the cross-entropy loss used for the pre-trained network approximates a Gaussian discriminant analysis classifier with a tied covariance matrix between classes [34]. To date, this simple metric approach has been the most successful for adapting a pre-trained classification network to perform OOD detection of outlier samples.

**2.1.3 One-class classification.** Another technique for determining a decision boundary in feature space to separate in-distribution data from outlier data is to attach a one-class classifier to the deep feature representation of a pre-trained network. The most popular one-class classification techniques are currently based on Support Vector Machines (SVM) [22, 35]. One-class SVMs find the maximum margin decision boundary such that some portion of training samples fall inside the boundary. The estimate of the proportion of training data that should be considered as the 'outlier' class is a hyper-parameter that must be set through cross-validation. Beyond SVMs, recent work has also focused on training deep one-class neural networks that learn an additional feature space to enable anomaly detection [36, 37]; however, training these auxilliary novelty detectors on top of a pre-trained CNN can be much more time consuming than many of the other methods mentioned here. For our study we evaluate the one-class SVM approach as a representative way of applying a one-class classifier for detecting novel samples in a pre-trained CNN.

**2.1.4 Extreme value theory.** OSC methods based on extreme value theory (EVT) recognize novel inputs by characterizing the probability of occurrences that are more extreme than any previously observed. This is typically implemented by characterizing class-conditional distributions in feature space. It has been directly adapted to CNN classifiers by modeling the distance to the nearest class mean in deep feature space as an extreme value distribution [38, 39] and calculating an acceptance score function as the posterior probability based on this EVT distribution. OpenMax [21] specifically applies EVT to construct a sample weighting function to re-adjust the output activations of a CNN based on a per-class Weibull probability distribution. The output is rebalanced between the closed set classes and a rejection class, and samples are rejected if the rejection class has a maximum activation or if the maximum activation falls below a threshold set from cross-fold validation.

## 2.2 Feature representation methods

In contrast to inference methods that calculate an acceptance score function from the feature space of a pre-trained network, we define feature representation OSC methods as strategies that alter the architecture of the network or how the network is trained in order to learn representations that enable better OSC performance.

**2.2.1 One-vs-rest classifiers.** The most common method for training a CNN classifier with $K$ disjoint categories is using cross-entropy loss calculated from a softmax activation function. Although the softmax function is good for training a classifier over a closed set of classes, it is problematic for outlier detection because the output probabilities are normalized, resulting in high-probability estimates for inputs that are either absurd or intentionally produced to fool a network [15, 23]. One-vs-rest classification models eliminate the softmax layer of a traditional closed-set classifier and replace it with a logistic sigmoid function for each class. While these per-class sigmoid activations no longer have a probabilistic interpretation in a multi-class problem, they reduce the risk of incorrectly classifying a novel sample by treating each class as a closed-set classification task, which can be individually thresholded to identify outliers. The aforementioned DOC model is one version of a one-vs-rest classifier that replaces the traditional softmax layer with a one-vs-rest layer of individual logistic sigmoid units [20].

**2.2.2 Background class regularization.** Another method for improving OSC performance via feature space regularization is using a representative background class to separate novel classes from known training samples. This technique is most commonly applied in object detection algorithms where the use of separate region proposal and image classification algorithms result in a classifier that must handle ambiguous object proposals [40]. Often these classifiers represent the background class as a separate output node which is trained using

datasets that have an explicit 'clutter' class such as MS COCO [41] or Caltech-256 [42]. Alternatively, newer approaches have used background samples to train a classifier to predict a uniform distribution when presented with anything other than an in-distribution training sample [43]. This is done by using an auxillary loss function that penalizes model predictions that diverge from a uniform distribution for novel samples. Recent methods that have incorporated this regularization for OSC include Confidence Loss [44] and Entropic Open Set Loss [45]. These methods have better OSC performance than just modeling the background class as an additional output node from the classifier. Nevertheless, evaluation of these methods on large-scale datasets has been very limited to date because of the difficulty in constructing a robust large-scale background dataset exclusisve of the training classes. We explore this shortcoming by evaluating models on large-scale datasets with various background datasets.

**2.2.3 Generative models.**   Using CNNs for generative modeling has been an active area of research with the advent of generative adversarial networks [46] and variational auto-encoders [47]. Generative models have extended earlier density estimation approaches for outlier detection by more accurately approximating the input distribution. A well-trained model can be used to directly predict if test samples are from the same input distribution [48] or estimate this by measuring reconstruction error [49]. Specifically to train open set classifiers, generative models have been used to create outlier inputs from the training set in order to condition a classifier to produce low confidence estimates, similar to how an explicit background class is used for model regularization [50]. These methods have proven less powerful than a well chosen background class for anything other than simple datasets [44, 51]. Thus, they are not explicitly evaluated in our comparisons.

## 3 Feature space visualization

To visually illustrate the differences between various methods, we trained a simple model for outlier detection using the MNIST dataset. We used a shallow CNN with a bottle-necked feature layer, i.e., the LeNet++ architecture [52], to allow visualization of the resulting decision boundaries. Fig 1 shows the 2-D decision boundaries with blue representing in-distribution classification at a 95% true positive rate threshold and red representing the resulting rejection region. Additionally, we mapped samples from an unknown class represented by the Fashion-MNIST [53] dataset in Fig 2 to understand how the decision boundaries relate to the deep CNN features of known and unknown classes.

These results illustrate that for a given feature space, inference strategies can be divided between those that have unbounded acceptance regions (e.g., $\tau$-Softmax) with those that are bounded (e.g., OpenMax). Much has been made of this distinction [3] and it is seen as a strength of the inference methods with bounded regions. However, as Fig 2 represents, unknown inputs are rarely mapped into these unbounded regions, but rather are centered around the origin in the deep feature space of a CNN. This implies that properly mapping the acceptance/rejection region around the origin is critical for OSC performance. Of the bounded acceptance region methods, OpenMax and Mahalanobis create the most compact decision boundaries. However, having compact boundaries may not be the best option when generalization to test inputs and unknown novel inputs is desired.

The goal of different feature space regularization strategies is to build robustness into the deep feature space by separating knowns from potential unknowns. While naively the One-vs-Rest training strategy appears to be a good solution by creating more compact class conditional distributions, the technique does not directly impact how features from unknown inputs will be mapped into the deep feature space. Instead we see that regularizing the model with a representation of the unknown class creates better separation between the known and unknown

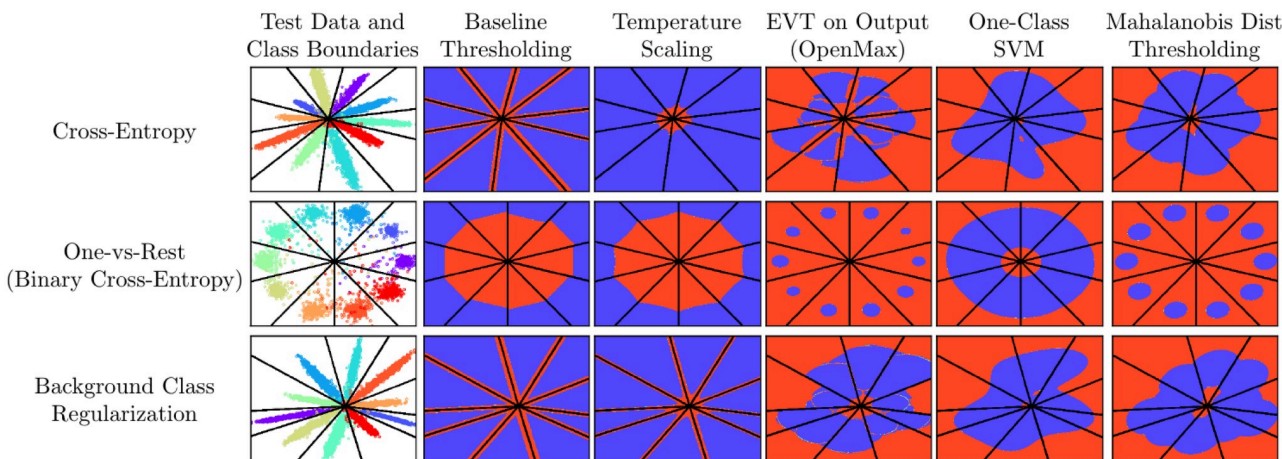

**Fig 1. Decision boundaries.** 2-D visualization of the decision boundaries created from the different inference methods studied using the LeNet + architecture and MNIST [52] as the training set. Blue is the acceptance region for in-distribution samples calibrated at a 95% True Positive Rate (TPR) for training data. Red is the rejection region (outlier).

[44, 45]. The difficulty in this approach, however, lies in large-scale datasets with many hundreds of classes.

## 4 Empirical assessment of open set classification methods

To compare performance of current state-of-the-art OSC methods on large-scale realistic datasets we asked three questions: 1) What is the benefit to be gained from the additional computational complexity that many of the specialized OSC inference methods introduce for detecting novel classes on complex large-scale visual recognition tasks using pre-trained ImageNet models? 2) How does does performance change as the similarity between the open set classes and closed set classes differs? Many of the methods tested have shown impressive results testing against OOD inputs drawn from either random noise or seperate datasets than the training set, but it is not clear whether performance degrades gradually or sharply as OOD samples become more similar to the in-distribution set, i.e., does the performance benefit from specialized inference methods increase or decrease as the open set detection problem becomes more

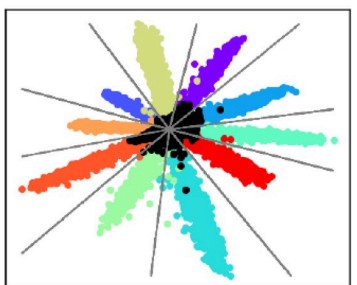 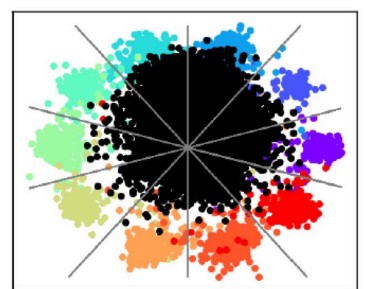 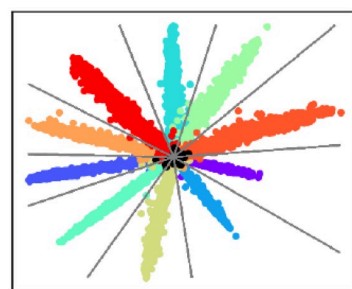

**(a)** Cross-Entropy  **(b)** One-vs-Rest  **(c)** Bkg. Class Reg.

**Fig 2. Feature space regularization.** 2-D visualization of the effect of the different feature space regularization strategies on separating in-distribution and outlier inputs The in-distribution training set is MNIST while the open set classes are drawn from the Fashion-MNIST dataset [53]. For background class regularization, the EMNIST-Letters dataset [54] is used as a source for background samples.

difficult. Finally, 3) Does additional model regularization during training aid in detection of open set classes when using one of the specialized inference mechanisms. Many of the aforementioned regularization techniques were only tested with baseline confidence thresholding to detect outlier samples, so it is unclear whether they would work harmoniously with current state-of-the-art inference methods.

To quantify OSC performance, we use two seperate metrics for evaluation. First we use the area under the receiver operating characteristic curve (AUROC) metric to assess OSC performance of each approach as a binary detector for seperating in-distribution and outlier samples. AUROC characterizes the performance across a full range of threshold values, regardless of the range of unique values for each inference method's scoring function. AUROC has been a commonly used metric for measuring OOD detection capabilities in image classification datasets [11, 12, 14, 43, 44]. This metric is best suited for comparing the inference methods which use the same pre-trained model; however, when comparing regularization techniques where the underlying closed-set accuracy can differ between models, we want a more discriminating measure.

Thus, we also adopt the area under the open set classification characteristic curve [45] (AUOSC), which is an adaptation on the traditional ROC curve measuring instead the correct classification rate versus false positive rate. This correct classification rate is the difference between the model accuracy and the false negative rate. Intuitively, this metric takes into account whether true positive samples are actually classified as the correct class and thus rewards methods which reject incorrectly classified positive samples before rejecting samples that are correctly classified. We extend the open set classification charactistic curve from [45] to calculate the area under the curve which provides an easy assessment of performance across different experimental paradigms and datasets.

To estimate the ability of OSC methods to scale, we trained models on the ImageNet large-scale image classification dataset (ImageNet). The ImageNet dataset was part of the ImageNet large-scale Visual Recognition Challenge [55] between 2012 and 2017 and evaluated an algorithm's ability to classify inputs into one of 1,000 possible categories. The dataset consists of 1.28 million training images (732-1300 per class) and 50,000 labeled validation images (50 per class), which we use for evaluation. We train an 18-layer ResNet model [1] for image classification on 500 randomly chosen classes, reserving the remaining 500 for intra-dataset OSC experiments.

To train the models, we use stochastic gradient descent with a mini-batch size of 256, momentum weighting of 0.9, and weight decay penalty factor of 0.0001. All models are trained for 90 epochs, starting with a learning rate of 0.1 that is decayed by a factor of 10 every 30 epochs. Training parameters were held constant for all feature space regularization strategies unless otherwise noted. The baseline cross-entropy trained model for the 500 class partition achieves 78.04% top-1 (94.10% top-5) accuracy.

## 4.1 Inference method comparison

To begin our assessment, we compare six of the inference methods described in Sec. 2 on large-scale image classification datasets using a pre-trained deep CNN model trained with cross entropy loss. The specific implementation details for the inference methods evaluated are as follows:

1. $\tau$-**Softmax**—This simple baseline approach finds a global threshold from the final output of the model after the associated activation function is applied. The method yields good results on common small-scale datasets [11] and can be easily extended to datasets with many classes.

2. **DOC**—Per-class thresholding has been shown to successfully reject outlier inputs during testing on common, small-scale datasets [56]. Adapting this method to larger datasets is more computationally expensive than $\tau$-Softmax because a per-class threshold must be established.

3. **ODIN**—This approach can outperform $\tau$-Softmax when using well-trained CNNs; however, the technique adds computational complexity during inference to calculate input perturbations [12]. ODIN also adds additional hyperparameters for the magnitude of input perturbation and a temperature scaling factor which must be determined through cross-validation.

4. **OpenMax**—OpenMax is one of the only methods previously tested on ImageNet [21]. It models a per-class EVT distribution and has multiple hyperparameters that must be tuned through cross-validation making it relatively cumbersome to use for large-scale datasets during training. Once these parameters have been found, however, it presents a robust inference method for estimating whether a sample belongs to one of the known classes or to an explicitly modeled outlier class. In the original implementation the decision rule for rejection involves a two step process where a sample is rejected as novel if either the outlier class is largest or the maximum class confidence of in-distribution classes is below a user-defined threshold. Because we evaluate methods across a range of thresholds, we have simplified this decision rule by setting the model confidence to zero if the outlier class is largest, otherwise the largest non-outlier confidence class is returned as the acceptance score value.

5. **One-class SVM**—One-class SVMs have been employed as a simple unsupervised alternative to density estimation for detecting anomalies. They have been tested across a wide variety of datasets, but not on the large-scale image datasets and CNN architectures used in this analysis. We use a radial basis function kernel to allow a non-linear decision boundary in deep feature space and tune hyperparameters via cross-validation.

6. **Mahalanobis**—In [14], the Mahalanobis metric was computed at multiple layers within a network and then combined via a linear classifier that was calibrated using a small validation set made up of in-distribution and outlier samples. To avoid biasing the model by training with open set data, we only compute the Mahalanobis metric in the final feature space. Adapting this metric to a large-scale dataset is straightforward, however, there is additional computational and memory overhead to estimate and store class conditional means and a global covariance matrix in feature space.

Hyperparameters for each inference method are tuned using outlier samples drawn from uniform noise to avoid unfairly biasing results to the datasets used for evaluation. Performance of each of the inference methods listed is shown in Fig 3.

Because the OSC performance increase from ODIN comes at a computational and memory cost, we also want to understand which elements of the inference method contribute most to the overall improvement in order to find the most efficient application of computational resources for OSC. To do this we performed an ablation of the method by looking at the OSC performance gained from temperature scaling and input perturbation independently as shown in Fig 4. For temperature scaling, we performed two variations: one where the temperature is chosen based on the procedure to minimize overall model calibration error as outlined in [18] and another where we grid search on a leave-out validation set for the best temperature for OSC performance independent of the impact on overall model calibration or closed set classification performance.

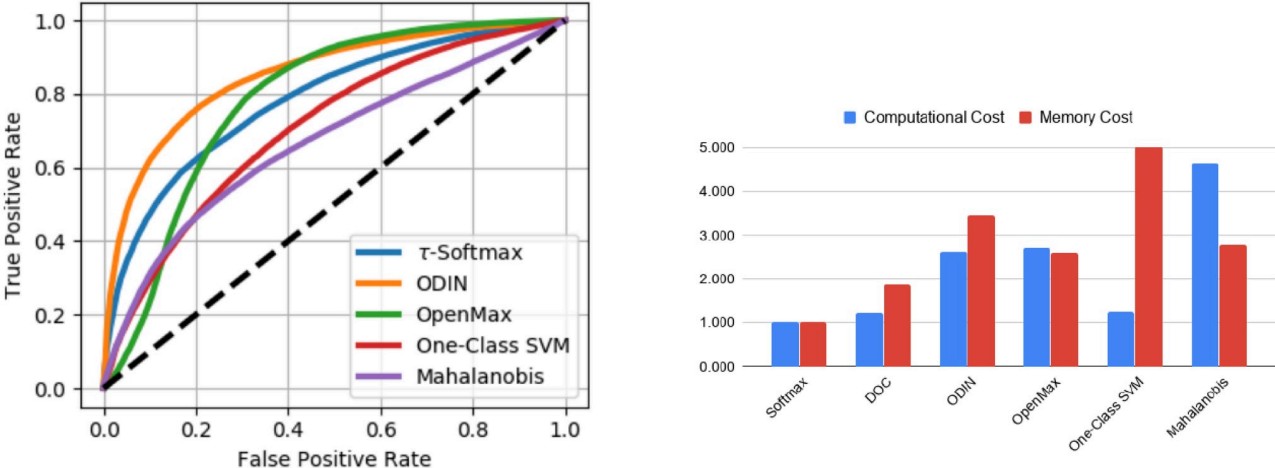

**Fig 3. Specialized inference methods for large-scale OSC.** Left) The ROC curve for various inference methods designed to determine whether samples are from known classes seen during training or from an unknown class. A ResNet-18 model is trained on a split made up of 500 randomly chosen categories from ImageNet. Evaluation is performed on the full ImageNet validation set with categories unseen during training being labeled as unknown (negative category). Right) The relative memory and computational cost of these specialized inference methods is shown relative to baseline confidence thresholding.

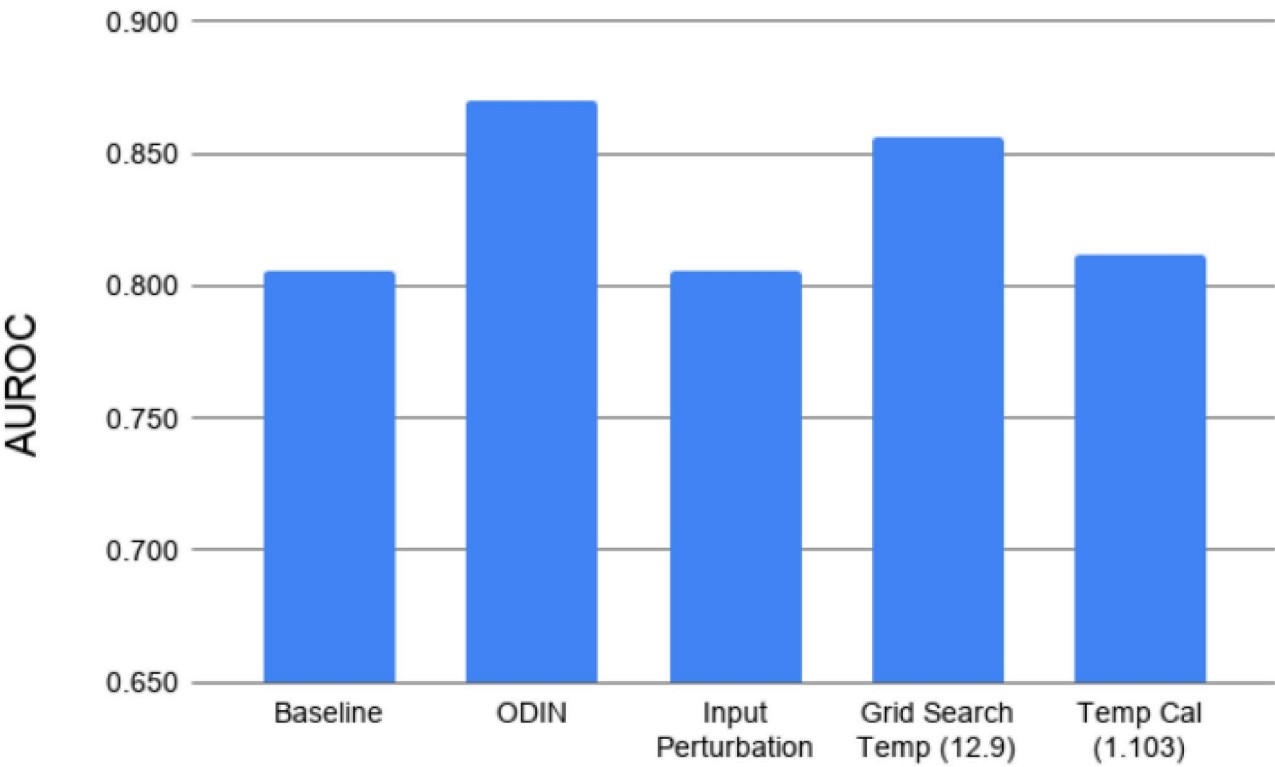

**Fig 4. Effects of input perturbation and temperature scaling.** An ablation experiment for detecting intra-dataset open set classes on a ResNet-18 model trained on 500 classes of ImageNet. As shown an optimal temperature scaling factor provides much of the overall performance benefit from the ODIN method with virtually no increase in computational complexity.

## 4.2 OSC similarity comparison

To evaluate the large-scale open set classification capabilities across a variety of OOD datasets, we created three separate outlier detection problems that vary in difficulty:

1. **Noise**: As the easiest OSC task, we evaluate both uniform and Gaussian noise inputs, which has been widely studied as a baseline [11, 12, 44, 57]. For the Gaussian images, we generate synthetic images from a zero mean, unit variance Gaussian distribution to match the data normalization scheme used for training our models.

2. **Inter-dataset**: As a problem of intermediate difficulty, we study each method's ability to detect outlier samples drawn from a seperate medium to high resolution image classification dataset. We include samples drawn from the Oxford Flowers dataset [58] and select categories of the Places dataset [59]. Specifically we removed overlapping categories from the Places dataset with our ImageNet training set determined using the hypernym/hyponym relationship from the Wordnet lexicon [60]. Additionally, for the Places dataset we sampled only from the outdoor categories leaving the indoor categories for regularization experiments as described below.

3. **Intra-dataset**: As the hardest outlier detection task, we used the remaining 500 categories from ImageNet that were not used for training.

In summary, the training set and models are kept fixed across the three paradigms, but the test sets vary across them. We construct the open set evaluation samples for each problem by randomly choosing 10,000 in-distribution samples evenly among the in-distribution classes and 10,000 outlier samples evenly among the open set classes within each respective dataset's validation set. This evaluation process was repeated 5 times and the resulting metrics were averaged.

To plot OSC performance across the varying OOD datasets against a meaningful metric, we use the maximum mean discrepancy (MMD) metric with a Gaussian kernel [61], i.e.,

$$\hat{MMD}^2(P, Q) = \frac{1}{N}\sum_{i \neq j} G(p_i, p_j) + \frac{1}{N}\sum_{i \neq j} G(q_i, q_j) - \frac{2}{N}\sum_{i \neq j} G(p_i, q_j)$$

where $P$ and $Q$ are the in-distribution and OOD sample spaces respectively and $G(\cdot, \cdot)$ is a Gaussian kernel whose scaling parameter is set to the median Euclidean distance of the aggregate set ($P \cup Q$). MMD is commonly used for quantifying the distance between datasets drawn from different distributions. Using this metric, we plot the OSC performance as a function of the MMD similarity in Fig 5.

## 4.3 Regularization comparison

Finally to assess the benefit of feature space regularization, we tested across three different training paradigms, including standard cross entropy training. The feature space regularization strategies for improving outlier detection were implemented as follows:

1. **Cross-entropy**—As a baseline, we train each network with standard cross-entropy loss to represent a common feature space for CNN-based models.

2. **One-vs-rest**—The one-vs-rest training strategy was implemented by substituting a sigmoid activation layer for the typical softmax activation and using a binary cross-entropy loss function. In this paradigm, every image is a negative example for every category it is not assigned to. This creates a much larger number of negative training examples for each class

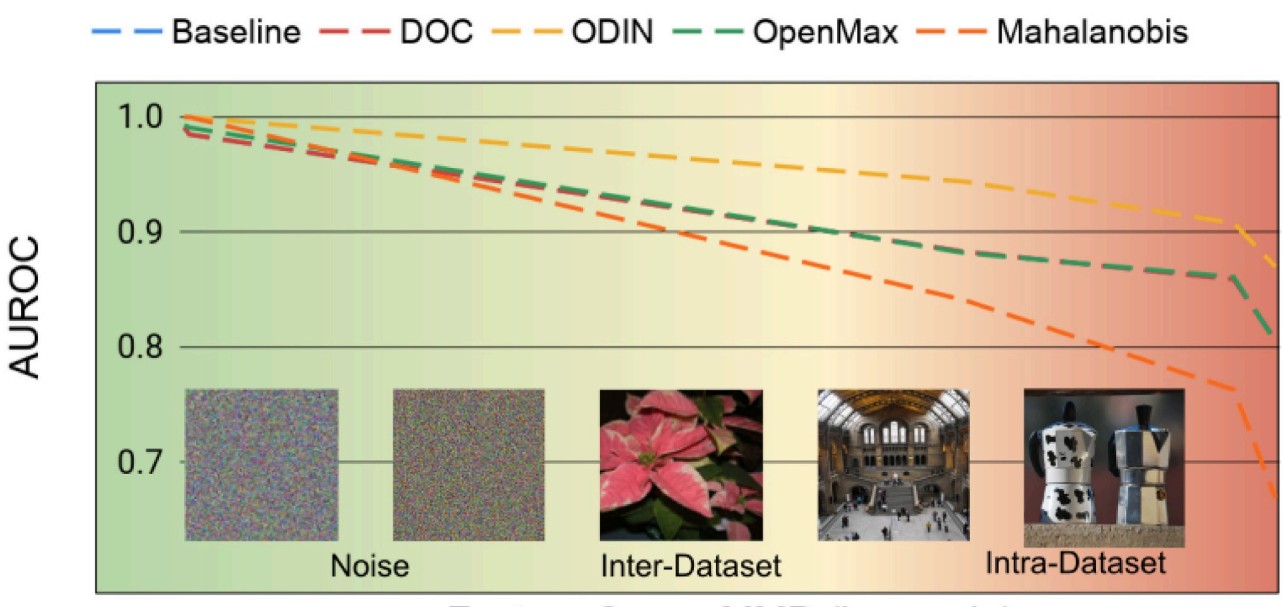

**Fig 5. OOD similarity comparison.** AUROC for OSC as a function of the MMD metric (log-scale, axis-reversed) measured from ResNet-18 embeddings for each dataset tested. For all methods tested, there is a large decrease in open set accuracy as the difference in feature representations of in-distribution and open set datasets decreases.

than positive examples. For this reason, we re-weight the negative-class training loss to be proportional to the positive-class loss to ensure comparable closed-set validation accuracy.

3. **Background class regularization**—The Entropic Open Set method [45] is a regularization scheme which uses a background class and a unique loss function during training to optimize the feature space of a neural network for separating known classes from potential unknowns. Similar to the confidence loss term in [44], the entropic open set loss forces samples from the background class to the null vector in feature space by calculating the cross-entropy of a uniform distribution for these samples. An additional regularization term is used to measure the hinge loss of the magnitude between samples in the background class and the training samples in feature space. For the background class, we use samples drawn from classes in the Places dataset that do not overlap with ImageNet classes and are distinct from the classes in the Places OSC experiments. Implementation details are described below.

For background class regularization of an ImageNet trained model, we use the Places dataset, which contains high-resolution images of scenes which are grouped into categories based on their human-related function [59]. We removed 103 categories from Places that overlapped with ImageNet, which were determined using the hypernym/hyponym relationship from the Wordnet lexicon [60]. The remaining classes were then split into outdoor and indoor subgroups. The indoor classes are used for training models that require background class regularization, while images from the outdoor classes are used for our inter-dataset OSC experiments of intermediate difficulty. Results from these experiments are shown in Table 2. In Fig 6 we also show the resulting ROC curves for the ImageNet Intra-Dataset problem across the three feature spaces tested. While qualitatively there appears to be little benefit from background class regularization versus standard cross-entropy training we did find significant differences

**Table 2. Area Under the Open Set Classification curve (AUOSC) for outlier detection and open set classification performance in ImageNet trained models averaged over 5 runs.** Top performer for each in-distribution / OOD combination is in blue along with statistically insignificant differences from the top performer as determined by DeLong's test [62] ($\alpha = 0.01$ with a correction for multiple comparisons within each column).

| Features Space | Inference Method | Gaussian | Places-Out | ImageNet-Open |
|---|---|---|---|---|
| CrossEntropy | $\tau$-Softmax | 0.786 | 0.713 | 0.688 |
| | DOC | 0.786 | 0.713 | 0.688 |
| | ODIN | 0.787 | 0.744 | 0.714 |
| | OpenMax | 0.786 | 0.712 | 0.687 |
| | One-Class SVM | 0.744 | 0.632 | 0.632 |
| | Mahalanobis | 0.751 | 0.523 | 0.502 |
| One-vs-Rest | $\tau$-Softmax | 0.649 | 0.539 | 0.539 |
| | DOC | 0.633 | 0.483 | 0.483 |
| | ODIN | 0.650 | 0.560 | 0.560 |
| | OpenMax | 0.649 | 0.500 | 0.500 |
| | One-Class SVM | 0.637 | 0.499 | 0.499 |
| | Mahalanobis | 0.623 | 0.439 | 0.439 |
| Background Class Regularization | $\tau$-Softmax | 0.751 | 0.746 | 0.717 |
| | DOC | 0.751 | 0.746 | 0.720 |
| | ODIN | 0.784 | 0.765 | 0.739 |
| | OpenMax | 0.734 | 0.672 | 0.737 |
| | One-Class SVM | 0.743 | 0.719 | 0.719 |
| | Mahalanobis | 0.750 | 0.545 | 0.493 |

in the AUROC metric calculated across the full range of OOD detection thresholds as reported in Table 2.

## 4.4 Model depth and width

Current state-of-the-art networks on large-scale image datasets often have hundreds of layers and hundreds of convolutional filters per layer. Previous work has shown that deeper and wider networks produce more accurate results, but often lead to uncalibrated predictions [18].

As an additional experiment, we desire to understand if there is a correlation between model capacity in a CNN, i.e., the depth and width of convolutional layers, and the resulting OSC performance. Our results indicate that in general novelty detection performance is related to overall model accuracy and varies as the feature space representation changes. To answer this question, we follow the protocol of [18] and train a series of ResNet models with either a fixed convolutional filter width (64) and varying depths (10-152 layers) or fixed depth (18 layers) and varying number of filter channels per layer (16-128). The results from these experiments are shown in Fig 7. Since performance on detecting open set samples largely tracks overall model accuracy it is not overly surprising that as the depth and width grow and model accuracy increases, then outlier detection performance also increases. Unlike the previously reported negative effect of model capacity on confidence calibration, there is no indication that increasing model depth or width negatively impacts OSC performance.

## 4.5 Discussion

For inference methods, we see that ODIN performs best on detecting open set classes from the ImageNet dataset for a pre-trained model across all three feature space regularization methods and across all outlier datasets evaluated. These results show the power of input perturbation and temperature scaling by showing improved performance over baseline methods and even

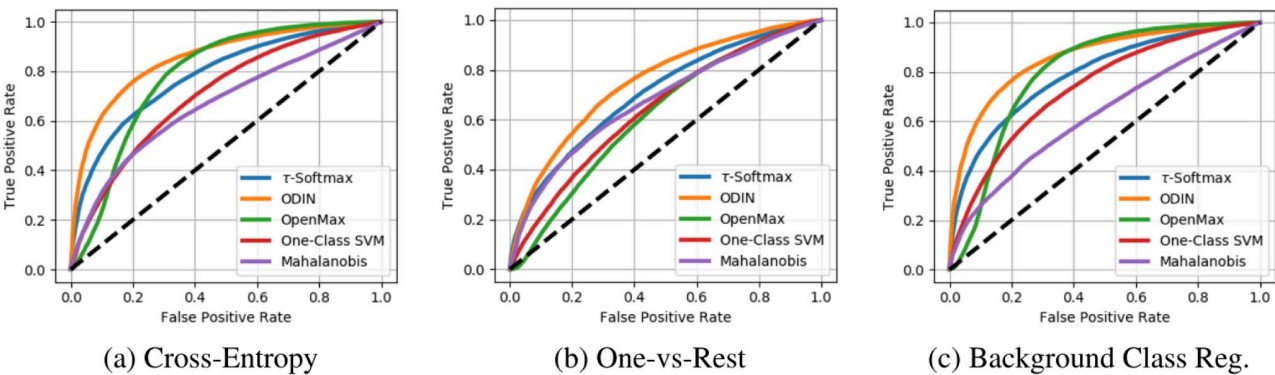

**Fig 6. Intra-dataset ROC curves.** Full ROC curves across the range of OOD detection thresholds for the Intra-Dataset (ImageNet-Open) experiment.

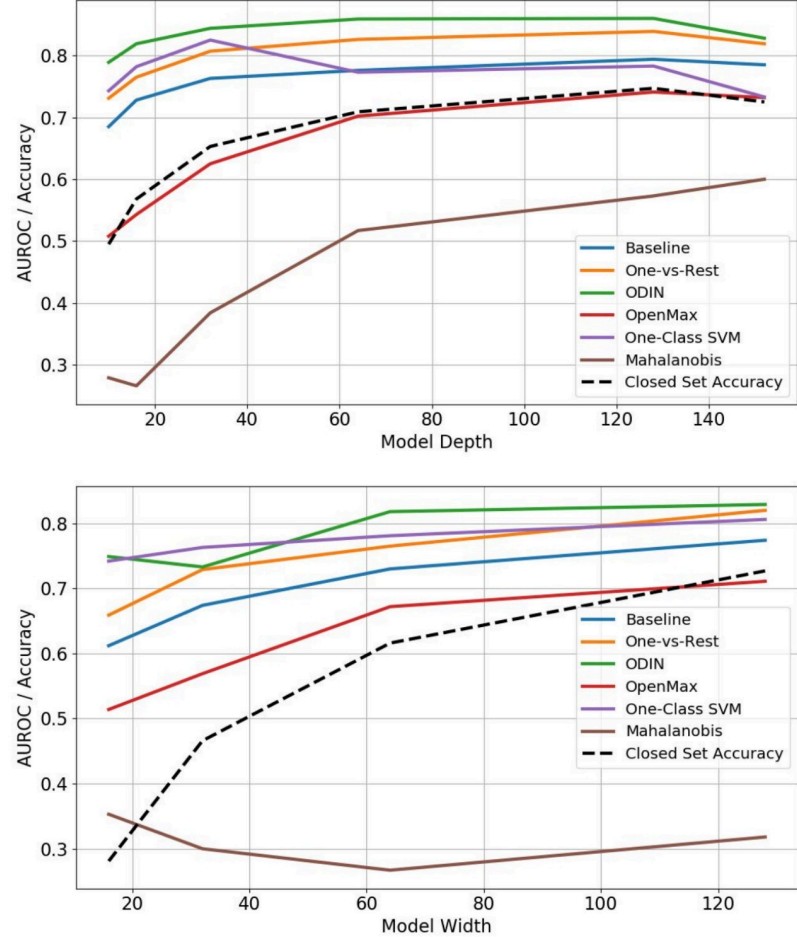

**Fig 7. OOD detection performance versus model capacity.** A ResNet architecture was varied in either depth or width and trained on the ImageNet-500 split and then tested for detecting image classes unseen during training via the remaining ImageNet categories (Intra-dataset). Overall improvements in performance as reflected in the AUROC of the model track improvements in model accuracy as model capacity increases.

more advanced methods on ImageNet, regardless of the feature space and difficulty of the OSC problem. However, this improvement comes at nearly a third reduction in image throughput and more than three times the memory cost during inference (see Fig 3). Results among the remaining inference methods are mixed, with the baseline global thresholding method ($\tau$-Softmax) performing equal to or better than all other methods for the most difficult, open set, outlier detection problem. Finally, while in general OSC performance decreases as the similarity between the OOD and in-distribution data increases, the relative performance increase of the ODIN method above the other methods tested is consistent across the different OOD datasets tested.

A large portion of the performance gain from ODIN can be acheived through temperature scaling alone as shown in Fig 4, which comes at virtually no increase in computational complexity or memory cost during inference as compared to the input perturbation method. This appears contradictory to recent improvements in OSC [13, 14] which have soley focused on utilizing the input perturbation methods to improve performance. Additionally our experiments show that finding a temperature scaling factor to optimize OSC performance is a separate task than finding the optimal temperature for minimizing calibration error.

Additionally, looking closer at the benefit of the different feature space representation methods tested, the results are mixed depending on the difficulty of the OSC problem. In general one-vs-rest training, results in reduced overall classification performance as seen in the lower AUOSC results which makes it a less desirable option for actually performing open set classification. Further, the benefit of background class regularization is demonstrated most significantly when detecting outlier samples that are similar to the background dataset used for training. The quantity of this improvement is reduced, however, as the OSC problem becomes more difficult. Nevertheless, background class regularization did not hurt either outlier detection or open set classification performance for any inference method except the Mahalanbois method.

Fundamentally, the increase in OSC difficulty as the similarity increases between OOD and in-distribution samples is due to the network confusing OOD inputs with known classes. This confusion stems from the feature space of the CNN classifier which learns to be most sensitive to variations in the training distribution that are semantically meaningful while ignoring variations that are not semantically meaningful among the known classes. Dealing with semantically meaningful variations in images from both known and unknown classes that are not included in the training set is ultimately the most significant problem in the OSC process.

## 5 Conclusion

Research in OSC has largely focused on either developing inference strategies for pre-trained models or a feature representation strategy for baseline inference methods for detecting outlier samples. However, as our results show, a large performance increase can be gained by combining an advanced inference technique with a feature space regularization strategy. Nevertheless, the performance increase over baseline techniques appears to be much smaller as the dataset becomes more complex and the novelty detection problem becomes more difficult.

In this paper, we performed a comprehensive comparison of OSC methods for CNNs using large-scale image classification datasets. We organized strategies into inference and feature space regularization methods, outlined the general applicability of these methods, and tested unique combinations of these two approaches previously unseen in any known literature. Additionally, we established a testing paradigm with varying difficulty using different outlier datasets. Through this paradigm, we demonstrated that novelty detection performance is very dataset dependent but generally decreases as the similarity between the in-distribution and

open set classes increases. Finally, there is still difficulty applying current state-of-the-art feature representation strategies for OSC to large-scale datasets that work in accordance with advanced inference methods. Ultimately, challenges remain in adapting open set classification methods for large-scale datasets and producing reliable recognition of novel inputs.

## Author Contributions

**Conceptualization:** Ryne Roady, Tyler L. Hayes, Ronald Kemker, Ayesha Gonzales, Christopher Kanan.

**Data curation:** Ryne Roady, Tyler L. Hayes, Ayesha Gonzales, Christopher Kanan.

**Formal analysis:** Ryne Roady, Tyler L. Hayes, Ayesha Gonzales, Christopher Kanan.

**Funding acquisition:** Christopher Kanan.

**Investigation:** Ryne Roady.

**Methodology:** Ryne Roady, Tyler L. Hayes, Ronald Kemker, Christopher Kanan.

**Resources:** Christopher Kanan.

**Software:** Ryne Roady.

**Visualization:** Ryne Roady.

**Writing – original draft:** Ryne Roady, Christopher Kanan.

**Writing – review & editing:** Ryne Roady, Tyler L. Hayes, Ronald Kemker, Christopher Kanan.

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
