## [Decision Letter · Decision Letter 0]

14 Jul 2020

PONE-D-20-03185

Are open set classification methods effective on large-scale datasets?

PLOS ONE

Dear Dr. Roady,

Thank you for submitting your manuscript to PLOS ONE. After careful consideration, we feel that it has merit but does not fully meet PLOS ONE’s publication criteria as it currently stands. Therefore, we invite you to submit a revised version of the manuscript that addresses the points raised during the review process.

We look forward to receiving your revised manuscript.

Kind regards,

Hao Sun

Academic Editor

PLOS ONE

Journal Requirements:

'TH and CK were supported during this work in part by DARPA/MTO (https://www.darpa.mil/about-us/offices/mto) Lifelong Learning Machines program [W911NF-18-2-0263] and AFOSR (https://www.wpafb.af.mil/afrl/afosr/) grant [FA9550-18-1-0121].  The funders had no role in study design, data collection and analysis, decision to publish, or preparation of the manuscript.'

We note that one or more of the authors are employed by a commercial company: Paige, New York

3. We note you have included a table to which you do not refer in the text of your manuscript.

Please ensure that you refer to Table 1 in your text; if accepted, production will need this reference to link the reader to the Table.

Reviewers' comments:

Reviewer's Responses to Questions

**Comments to the Author**

1. Is the manuscript technically sound, and do the data support the conclusions?

Reviewer #1: Yes

Reviewer #2: Yes

2. Has the statistical analysis been performed appropriately and rigorously? 

Reviewer #1: Yes

Reviewer #2: Yes

3. Have the authors made all data underlying the findings in their manuscript fully available?

Reviewer #1: No

Reviewer #2: Yes

4. Is the manuscript presented in an intelligible fashion and written in standard English?

Reviewer #1: Yes

Reviewer #2: Yes

5. Review Comments to the Author

Reviewer #1: This paper studies the performance of major inference methods and regularization strategies for the purpose of open set classification(OSC). The manuscript is well written, comprehensively summarizing methods in different categories. It conducts multiple numerical tests using large-scale datasets and compares accuracy and computational costs of different methods. In the end, it concludes with some useful recommendations for researchers in this field.

I recommend accepting this work after the authors address some minor issues:

1. While this paper targets large-scale dataset, the studied ResNet only has 18 layers contrast to popular networks that have hundreds of layers and convolutional filters(Lines 433-434). Therefore, the authors further studies the effects of model depth and width in Section 4.4. However, in Figure 7 the authors only consider τ-Softmax and ODIN. Could the authors justify why other inference/regularization methods are not considered?

2. In Lines 461-463, the authors state that "in general OSC performance decreases as the similarity between the OOD and in-distribution data increases". Could the authors explain this observation?

3. In Lines 428-431, the authors says the resulting ROC curves "demonstrate that there is little to no benefit from background class regularization versus standard cross-entropy training in the open set classification task". However, in Table 2 it seems that the AUOSC values from background class regularization in ImageNet-Open do have some improvement versus the AUOSC values in cross-entropy. Additionally, in previous section the authors mention that AUOSC metric is a better indicator when comparing regularization techniques. Therefore, I feel that the authors should make a more comprehensive conclusion by considering Table 2 and Figure 6 together.

4. In Figure 7, it seems that the legends don’t tell which curves are from Intra-Dataset.

Reviewer #2: This paper presents a method to optimize the training/validation data distribution in supervised classification. The authors of the paper consider two approaches: 1 to separate knowns from unknown data and 2 feature space regularization strategies to improve model robustness to novel inputs. Different from traditional methods that focus on those approaches separately, the authors of the paper uniquely combine those two methods by exploring the relationship between the two approaches and directly comparing performance on ImageNet dataset. The authors consider regularization and specialized inference methods together for data augmentation, and find good result on large-scale dataset such as imagenet.

Advantages:

1. the authors of the paper present detailed elaboration on the problem, which is the performance of the open set classification in large-scale dataset.

2. the authors of the paper compare methods across open set classification paradigms on large-scale, high-resolution instead of low-resolution MNIST and CIFAR.

3. the authors of the paper compare inference methods and feature space regularization strategies and combine them for further evaluation for out-of-distribution problem.

4. the authors of the paper provide certain regularization method with better performance for out-of-distribution problem.

5. the experimental data are well organized.

6. the comparison baselines are enough and up-to-date.

Weakness:

1. I don't know if it is the problem with the latex template but why all figures are at the bottom of the paper? Please re-organize it if it is not the template matter.

2. I think the authors should consider to add more details on ablation study to this work, considering independent input perturbation and temperature scaling factors are all well studied in different works.

3. Input perturbation is essentially a data augmentation method that has been widely used in improving DNN performances. The authors of the paper should considering more data augmentation methods in their approach and evaluate which one is better. In the same time, this will also benefit the ablation study in Figure 4.

6. PLOS authors have the option to publish the peer review history of their article (what does this mean?). If published, this will include your full peer review and any attached files.

Reviewer #1: No

Reviewer #2: No

---

## [Author Response · Author response to Decision Letter 0]

12 Aug 2020

Responses to Reviewers and Revisions to the Paper

Reviewer #1: While this paper targets large-scale dataset, the studied ResNet only has 18 layers contrast to popular networks that have hundreds of layers and convolutional filters(Lines 433-434). Therefore, the authors further studies the effects of model depth and width in Section 4.4. However, in Figure 7 the authors only consider τ-Softmax and ODIN. Could the authors justify why other inference/regularization methods are not considered?

ODIN was chosen because it was one of the best performing methods, and τ-Softmax was provided as a comparison. We agree that other inference methods can easily be added and serve as additional evidence to our initial analysis that OSC performance generally follows the same trend as closed-set accuracy when the model capacity is varied within a ResNet architecture. We have updated Figure 7 to include the other inference methods in considered in our paper. Additionally we have concluded that including both the Intra-dataset and Inter-dataset data for this ablation is unnecessary as the trends from the inter-dataset (ImageNet-Open) OOD data is the most significant to identifying trends. 

Reviewer #1: In Lines 461-463, the authors state that "in general OSC performance decreases as the similarity between the OOD and in-distribution data increases". Could the authors explain this observation?

This was an empirical observation drawn from Figure 5. The underlying reasons for this performance decrease is due to OOD samples from novel classes being confused for known classes. This is likely due to the network learning features to distinguish between classes during normal discriminative training; however, if an OOD image shares some of these distinguishing features with a known class then there is a higher likelihood that the OOD image will be incorrectly identified. 

We added this hypothesis in the discussion section following the discussion of feature space regularization strategies:

Fundamentally, the increase in OSC difficulty as the similarity increases between OOD and in-distribution samples is due to the network confusing OOD inputs with known classes. This confusion stems from the feature space of the CNN classifier which learns to be most sensitive to variations in the training distribution that are semantically meaningful while ignoring variations that are not semantically meaningful among the known classes. Dealing with semantically meaningful variations in images from both known and unknown classes that are not included in the training set is ultimately the most significant problem in the OSC process. 

Reviewer #1: In Lines 428-431, the authors says the resulting ROC curves "demonstrate that there is little to no benefit from background class regularization versus standard cross-entropy training in the open set classification task". However, in Table 2 it seems that the AUOSC values from background class regularization in ImageNet-Open do have some improvement versus the AUOSC values in cross-entropy. Additionally, in previous section the authors mention that AUOSC metric is a better indicator when comparing regularization techniques. Therefore, I feel that the authors should make a more comprehensive conclusion by considering Table 2 and Figure 6 together.

We have tempered this statement in describing the lack of benefit from background regularization in large-scale OSC problems. We have added the additional observation that while the ROC charts qualitatively appear to show little benefit from the background class regularization, we have nevertheless shown statistically significant increases in AUROC performance from this feature space regularization approach. The specific wording of the paragraph has been changed to:

In Fig. 6 we also show the resulting ROC curves for the ImageNet Intra-Dataset problem across the three feature spaces tested. While qualitatively there appears to be little benefit from background class regularization versus standard cross-entropy training we did find significant differences in the AUROC metric calculated across the full range of OOD detection thresholds as reported in Table 2.

Reviewer #1: In Figure 7, it seems that the legends don’t tell which curves are from Intra-Dataset.

We have updated Figure 7. It now includes a wider variety of inference methods as explained in the first response above.

Reviewer #2: I don't know if it is the problem with the latex template but why all figures are at the bottom of the paper? Please reorganize it if it is not the template matter.

The submission template for PLOS ONE required figures to be submitted separately from text. 

Reviewer #2: I think the authors should consider to add more details on ablation study to this work, considering independent input perturbation and temperature scaling factors are all well studied in different works. Input perturbation is essentially a data augmentation method that has been widely used in improving DNN performances. The authors of the paper should considering more data augmentation methods in their approach and evaluate which one is better.

We have included in our ablation of the ODIN method an independent analysis of input perturbation and temperature scaling. While we believe different data augmentation methods will have a significant effect on OOD detection performance, we focused on using the standard methods used by the creators of the approaches we are comparing to assess their capabilities with different inference methods on large-scale datasets.

---

## [Editor Report · Decision Letter 1]

14 Aug 2020

Are open set classification methods effective on large-scale datasets?

PONE-D-20-03185R1

Dear Dr. Roady,

We’re pleased to inform you that your manuscript has been judged scientifically suitable for publication and will be formally accepted for publication once it meets all outstanding technical requirements.

Kind regards,

Hao Sun

Academic Editor

PLOS ONE

---

## [Editor Report · Acceptance letter]

26 Aug 2020

PONE-D-20-03185R1 

Are open set classification methods effective on large-scaledatasets? 

Dear Dr. Roady:

I'm pleased to inform you that your manuscript has been deemed suitable for publication in PLOS ONE. Congratulations! Your manuscript is now with our production department. 

Kind regards, 

on behalf of

Professor Hao Sun 

Academic Editor

PLOS ONE